# MiRNA Profiling and Its Potential Roles in Rapid Growth of Velvet Antler in Gansu Red Deer (*Cervus elaphus kansuensis*)

**DOI:** 10.3390/genes14020424

**Published:** 2023-02-07

**Authors:** Zhenxiang Zhang, Caixia He, Changhong Bao, Zhaonan Li, Wenjie Jin, Changzhong Li, Yanxia Chen

**Affiliations:** 1Qinghai Provincial Key Laboratory of Adaptive Management on Alpine Grassland, Academy of Animal Science and Veterinary Medicine, Qinghai University, Xining 810016, China; 2College of Eco–Environmental Engineering, Qinghai University, Xining 810016, China

**Keywords:** *Cervus elaphus kansuensis*, velvet antler, growth center, rapid growth, microRNAs

## Abstract

A significant variety of cell growth factors are involved in the regulation of antler growth, and the fast proliferation and differentiation of various tissue cells occur during the yearly regeneration of deer antlers. The unique development process of velvet antlers has potential application value in many fields of biomedical research. Among them, the nature of cartilage tissue and the rapid growth and development process make deer antler a model for studying cartilage tissue development or rapid repair of damage. However, the molecular mechanisms underlying the rapid growth of antlers are still not well studied. MicroRNAs are ubiquitous in animals and have a wide range of biological functions. In this study, we used high-throughput sequencing technology to analyze the miRNA expression patterns of antler growth centers at three distinct growth phases, 30, 60, and 90 days following the abscission of the antler base, in order to determine the regulatory function of miRNA on the rapid growth of antlers. Then, we identified the miRNAs that were differentially expressed at various growth stages and annotated the functions of their target genes. The results showed that 4319, 4640, and 4520 miRNAs were found in antler growth centers during the three growth periods. To further identify the essential miRNAs that could regulate fast antler development, five differentially expressed miRNAs (DEMs) were screened, and the functions of their target genes were annotated. The results of KEGG pathway annotation revealed that the target genes of the five DEMs were significantly annotated to the “Wnt signaling pathway”, “PI3K-Akt signaling pathway”, “MAPK signaling pathway”, and “TGF-β signaling pathway”, which were associated with the rapid growth of velvet antlers. Therefore, the five chosen miRNAs, particularly ppy-miR-1, mmu-miR-200b-3p, and novel miR-94, may play crucial roles in rapid antler growth in summer.

## 1. Introduction

The appendages of male deer, velvet antlers, are rich in blood vessels and nerves and covered with velvet skin [1]. It is the only appendage organ in mammals which can regenerate every year [2]. Androgen-regulated antlers undergo annual cyclic shedding and regrowth, which is distinct from cellular dedifferentiation in lower organisms but based on the differentiation process in stem cells [3]. Antler’s growth is a very complex process. During rapid growth, the cells proliferate rapidly but remain organized and not neoplastic. The antlers eventually undergo complete ossification, resulting in the death of velvet antler tissue and effectively preventing the uncontrolled growth of the antler [4]. However, as an ideal growth system model for cartilage damage repair and cell proliferation and differentiation without carcinogenesis, the mechanism of antler rapid growth remain unclear.

The growing season of antlers varies from being slow in spring to being exponentially accelerated in summer to being slow in autumn with a typical S curve [5]. In summer, antlers grow fast and are covered in velvet skin. In autumn, due to the rapid increase in the androgen level in deer, antlers are rapidly ossified, and the velvet skin is gradually dried up and shed. In winter, completely ossified and dead antlers are tightly attached to the living tissue antler pedicle. The following spring, the antlers fall off, and a new cycle of antler regeneration begins [6]. The whole process of antler growth and development is divided into anagenesis and ossification, with the regeneration of new blood vessels and nerves. The morphology of an antler consists of anagenesis between a tiny saddle (30 days) and a three-branched antler (90 days), which is mostly made up of growth and a sluggish ossification level. Between these two points, the growth of a two-bar antler (60 days) represents a period of rapid growth, with a growth rate of up to 2 cm/d. When the three-branched antler enters the ossification period, the growth rate decreases, and the ossification level increases [6]. The rapid growth of an antler is closely related to the proliferation, differentiation, and hypertrophy of apical chondrocytes, with the abundant blood supply in the pre-cartilage and cartilage areas and a higher level of angiogenesis in the pre-cartilage area [7,8]. Antler growth is primarily driven and controlled by the apical growth center through cartilaginous internalization [1]. The growth center of an antler mainly includes the dermis, mesenchyme, and cartilage tissue [9,10]. The apical tissues of velvet antlers have been the subjects of research to understand better the biological mechanisms underlying the rapid growth of antlers and to discover important genes [11,12,13].

MicroRNAs are endogenous small single-stranded non-coding RNAs generated by the transcriptional processing of non-coding regions of eukaryotic genomes. They are abundantly distributed across all eukaryotes, including unicellular algae, and their structural details have been remarkably conserved throughout biological history [14]. After extensive processing and maturation, microRNAs are around 21–25 nt in length and bind to the mRNA 3′ non-coding region (3′-UTR) of target genes in a totally or partially complementary way, then affecting the expression of the target genes at the transcriptional or post-transcriptional level [15]. MicroRNAs can inhibit the translation of their target mRNAs without affecting mRNA transcription [16]. They play a crucial role in the formation of the body, organogenesis, a variety of illnesses, and other processes and are potent regulators of numerous cellular functions, such as cell proliferation, differentiation, development, and apoptosis [17,18]. Additionally, research in the field of deer antler biology has demonstrated that velvet antler growth and development can be influenced by miRNAs either directly or indirectly [19,20,21,22,23,24].

At present, the molecular mechanism of the rapid growth of velvet antlers has not been studied in detail. In this study, the miRNA expression at different growth stages in Gansu Red deer antler growth centers was studied via high-throughput sequencing. By predicting and analyzing the functions of miRNA target genes, it was possible to explore the mechanism of miRNAs participation in the growth and development of antlers. With these initiatives, we hope to learn more about miRNA expression patterns in antler growth and develop a theoretical framework for future research regarding the molecular mechanisms of miRNAs in the rapid growth and regeneration of antlers.

## 2. Materials and Methods

### 2.1. Sample Collection

Antlers of three healthy adult individuals (4–5 years old) from the Shandan Farm in Gansu Province, China, were collected for this study. At different growth stages (30, 60, and 90 days), the apical (~5 cm) portions of the antlers were harvested and cut into thin slices which were approximately 2 mm longitudinally along the sagittal plane, and then, the antler growth centers (dermis, mesenchyme, and anterior cartilage layer) were separated according to Chunyi Li’s method [9]. In each phase, samples were taken from three different individuals. The collected samples were cleaned with PBS, frozen with liquid nitrogen, and brought back to the lab to be stored at −80 °C for RNA extraction. After the deer were anesthetized, the antlers were collected, and once they were cut, hemostatic procedures were promptly carried out. After the deer awoke, they were returned to the deer herd.

### 2.2. RNA Extraction, Library Preparation, and Sequencing

Total RNA was extracted using the TRIzol reagent (Thermo Fisher Scientific, Waltham, MA, USA) in accordance with the manufacturer’s instructions. DNase I (Promega, Madison, WI, USA) was used to clean up extracted RNA and remove contaminants. The concentration and integrity of the RNA samples were assessed using a NanoDrop 2000 spectrophotometer (Thermo Fisher Scientific, USA), RNA Nano 6000 analysis kit, and Bioanalyzer 2100 system (Agilent Technologies, Santa Clara, CA, USA) to confirm that they complied with sequencing requirements. It was found that the extracted total RNA satisfied the quality standards and could therefore be utilized for the construction of small RNA libraries if 1.8 < OD260/OD280 < 2.0, RIN > 7.0, and rRNA 28S/18S ≥ 0.7. Utilizing the NEBNext^®^ Ultra™ RNA Library Prep Kit for Illumina^®^ (New England Biolabs, Ipswich, MA, USA), sequencing libraries were created. A Bioanalyzer 2100 (Agilent Technologies) was used to test the quality of the libraries. The miRNA libraries were sequenced using the HiSeq™ 2500 platform (Illumina, USA).

### 2.3. Data Processing and Analysis

By eliminating spliced sequences, reads with poly(A) tails, low-quality bases, and sequences with less than 18 or more than 30 nucleotides, raw data were initially processed to obtain clean reads. Additionally, the cleaned dataset’s Q20 and Q30 scores, GC content (%), and levels of sequence repetitions were calculated. The Trinity software [25] was used to carry out de novo assembly. Secondly, clean reads were aligned with Silva (https://www.arb-silva.de, accessed on 15 September 2022), genomic tRNA (GtRNAdb; http://lowelab.ucsc.edu/GtRNAdb/, accessed on 15 September 2022), Rfam (http://rfam.xfam.org/, accessed on 16 September 2022), and Repbase (www.girinst.org/Repbase, accessed on 16 September 2022) databases using Bowtie to remove sequences matching rRNA, tRNA, SNRNA, SnRNA, other NCRNA, and repetitive sequences. Thirdly, the secondary structures of novel miRNAs were predicted using the Randfold program (http://www.aquafold.com, accessed on 19 September 2022). Fourth, the known miRNAs were determined via the miRBase v21.0 database (http://www.mirbase.org/, accessed on 20 September 2022) using either 0 or 1 mismatches. The novel miRNAs were predicted using RNAfold software [26] for sequences that could not be aligned to the miRBase database but could be mapped to the red deer genome. Expression profiles of miRNA were computed using the normalized approach [27]. DEseq software [28] was used to identify miRNAs that were differentially expressed across the three libraries based on normalized counts. The differentially expressed miRNAs (DEMs) were detected using |log2(Fold Change, FC)| ≥ 1.00 and False Discovery Rate (FDR) ≤ 0.01 as the screening criteria. Then, data from the three sample groups were compared in pairs (d30 versus d60, d30 versus d90, and d60 versus d90), and the DEMs of each comparison group were screened, respectively.

We used the default settings of the software programs Miranda v33 (http://www.microrna.org, accessed on 25 September 2022) and TargetScan 7.2 (http://www.targetscan.org, accessed on 25 September 2022) to estimate possible target genes of miRNAs. Both approaches’ predictions of genes were taken into consideration when choosing possible targets for certain miRNAs. The Kyoto Encyclopedia of Genes and Genomes (KEGG) pathway analysis and GO enrichment (Gene ontology) were used to annotate these candidate genes with default parameters functionally.

### 2.4. Real-Time PCR to Verify Differentially Expressed microRNAs

The DEMs in the antler growth centers of Gansu red deer during the three growth periods were detected via qPCR. The *U6* gene was used as the internal reference, and the primers of the DEMs and *U6* gene are listed in Table 1. RNA was extracted from antler growth centers using the TaKaRa MiniBEST Universal RNA Extraction Kit (TaKaRa, Code No. 9767), and 1 ug of total RNA from each sample was reverse transcribed to synthesize cDNA using PrimeScript™ RT Master Mix (Perfect Real Time) (TaKaRa, Code No. RR036Q), which used as the template for qPCR. qPCR was performed with TB Green^®^ Premix Ex Taq™ II (Tli RNaseH Plus) (TaKaRa, Code No. RR820A). qPCR reaction system (20 μL): 2 μL of cDNA, 1 μL of forward primer (10 μM), 1 μL of reverse primer (10 μM), 10 μL of TB Green Premix Ex Taq II (Tli RNaseH Plus) (2×), and 6 μL of ddH_2_O. qPCR reaction conditions: pre-denaturation at 95 °C for 10 min; denaturation at 95 °C for 15 s, and annealing at 60 °C for 30 s for 40 cycles; extension at 72 °C for 5 min; storage at 4 °C. The relative expression results were calculated and analyzed via the 2^−ΔΔCt^ method.

## 3. Results

### 3.1. Sequencing Data Processing

The sequencing results showed that 14,971,388, 16,985,192, and 15,732,412 raw reads were obtained from the 30-, 60-, and 90-day growth centers of red deer velvet antlers, respectively. Further processing of the raw data resulted in 9,203,007, 12,661,360, and 10,346,245 clean reads in three groups (d30, d60, and d90), with Q30 percentages higher than 98.00% for each sample. After removing rRNA, tRNA, SNRNA, SnRNA, other NCRNAs, and repeat sequences, 8,083,271, 11,313,839, and 9,089,417 unannotated reads from the three groups were compared to the red deer reference genome; then 5,047,174, 6,834,924, and 5,384,420 mapped reads were obtained in the three groups, which accounted for 52.09%, 45.12%, and 46.46% of their unannotated reads, respectively. Finally, the mapped reads were aligned with the miRBase database for miRNA identification. Based on these results, the clean reads were reliable, and data quality was consistent and good, which meant that it could be used for subsequent analysis.

### 3.2. MiRNA Identification

The mapped reads were aligned with the miRBase database for the identification of conserved miRNAs, and miRDeep2 software was used for the prediction of novel miRNAs. In total, 4103, 4429, and 4307 known miRNAs were identified, and 216, 211, and 213 novel miRNAs were predicted in each of the three groups. The length distribution of the identified known miRNAs and new miRNAs showed that the miRNAs detected in this study were 18–28 bp in length, and most of them were concentrated at 20–24 bp, with 22 bp being predominant (Figure 1). The expression abundance data were generated by homogenizing miRNA expression across different samples; Table 2 and Table 3 show the top 20 known and novel miRNAs with the highest levels of expression in all three groups. Among all the identified miRNAs, 4136 were expressed in d30, d60, and d90, and 41, 161, and 91 miRNAs were specifically expressed in groups d30, d60, and d90, respectively (Figure 2).

### 3.3. Screening for Differentially Expressed miRNAs

The miRNAs that were differentially expressed in the different comparison groups (d30 vs. d60, d90 vs. d60, and d30 vs. d90) were filtered out. A total of 292, 102, and 60 miRNAs were found to be differentially expressed in the three comparison groups, respectively. In total, 122 DEMs were significantly up-regulated in d30 compared to the expression of miRNAs in d60, and 170 DEMs were significantly down-regulated. Compared with the expression of miRNA in d90, 38 DEMs were significantly up-regulated in d30, 64 DEMs were significantly down-regulated in d30; the expression of 40 DEMs was significantly up-regulated in d60, and 20 DEMs were significantly down-regulated in d60 (Figure 3, Appendix A).

### 3.4. Target Genes’ Prediction of Differentially Expressed miRNAs

The target genes of DEMs in the three comparison groups were predicted using two algorithms, TargetScan and miRanda. A total of 14,050 target genes were identified after the intersection, of which 13,086 genes were predicted for conserved miRNAs, and 9584 genes were predicted for novel miRNAs. Specifically, 13,162, 13,807, and 13,725 target genes were obtained in antler growth centers at days 30, 60, and 90, respectively.

In terms of differentially expressed miRNAs, there were 3419, 3719, and 3665 in the d30 vs. d60, d30 vs. d90, and d60 vs. d90 comparison groups, respectively. Specifically, in the 30d vs. 60d comparison group, up-regulated DEMs predicted 1189 target genes, and down-regulated DEMs predicted 2663 target genes. In the d30 vs. d90 comparison group, up-regulated DEMs predicted 3308 target genes, and down-regulated DEMs predicted 730 target genes. In the d60 vs. d90 comparison group, up-regulated DEMs predicted 3603 target genes, and down-regulated DEMs predicted 115 target genes.

### 3.5. Target Genes’ Functional Analysis of Differentially Expressed miRNAs

We examined the roles played by the target genes of the DEMs to better understand the roles of miRNA target genes in antler growth centers during various growth phases. The results of the GO enrichment analysis showed that the target genes of differentially expressed miRNAs were significantly enriched in 30 GO categories (*p* < 0.05) in all three comparison groups, including “cellular process”, “biological regulation”, and “single-organism process” in Biological Process (BP); “cell part”, “cell”, and “organelle” in Cellular Component (CC); and “binding” and “catalytic activity” in Molecular Function (MF) (Figure 4).

The KEGG pathway enrichment analysis revealed that in the d30 vs. d60 comparison group, the functions of the target genes of DEMs were significantly enriched in 232 KEGG pathways, including “Type II diabetes mellitus”, “Axon guidance”, and “Small cell lung cancer”, among others. The target genes of DEMs were enriched in 230 KEGG pathways in the comparison group between d30 and d90, with target genes significantly enriched in “Basal cell carcinoma”, “Axon guidance”, “Endocytosis”, etc. In the d60 vs. d90 comparison group, differential target genes were significantly enriched in “Basal cell carcinoma”, “Endocytosis”, “Axon guidance”, etc. (*p* < 0.05). Figure 5 shows the top 20 signaling pathways with the highest confidence of enrichment significance (i.e., the smallest Q value) across the three comparison groups.

Additionally, we were intrigued by the molecular processes that underlining the quick development of velvet antlers in the rapid growth stage (about 60 days). As miRNAs usually negatively regulate the expression of their target genes, in this study, we screened four DEMs that were lowly expressed in antler growth centers at 60 days and highly expressed in antler growth center tissues at 30 and 90 days, including ppy-miR-1, mmu-miR-200b-3p, novel miR-6, and novel miR-94. In addition, there was another miRNA (novel miR-10) with expression characteristics that were opposite to those of the above four miRNAs. Target gene prediction results showed that ppy-miR-1, mmu-miR-200b-3p, novel miR-6, novel miR-94, and novel miR-10 predicted 821, 1154, 121, 458, and 737 target genes, respectively. The results of the KEGG pathway annotation showed that the target genes of ppy-miR-1, mmu-miR-200b-3p, novel miR-6, novel miR-94, and novel miR-10 were significantly annotated to “Ras signaling pathway”, “Neurotrophin signaling pathway”, “Glioma”, “Pathways in cancer”, and “Thyroid hormone signaling pathway” (*p* < 0.05), respectively. Furthermore, the target genes of these miRNAs were also annotated to the “Wnt signaling pathway”, “PI3K-Akt signaling pathway”, “MAPK signaling pathway”, “TGF-β signaling pathway”, and “mTOR signaling pathway”, and other classical signaling pathways (Appendix A). It is possible that these miRNAs may play important roles in the rapid growth of velvet antlers, which needs to be further verified.

### 3.6. Validation of miRNAs Expression

The Venn Diagram package of R language was used to analyze the differential miRNAs, and there were 13 common DEMs in the three comparison groups. After removing the miRNAs with consistent, mature sequences, three DEMs (bta-miR-1298, bta-miR-486, and novel miR-166) were retained for verification via stem-loop RT-qPCR. In addition, to further investigate the function of the four DEMs (ppy-miR-1, mmu-miR-200b-3p, novel miR-6, and novel miR-94), their relative expression in the growth centers of velvet antlers in three growth stages was also detected.

The expression levels of bta-miR-1298 and novel miR-166 increased with antler growth, and the expression levels of bta-miR-486 showed a rising then falling trend, while the expression levels of ppy-miR-1, mmu-miR-200b-3p, novel miR-6, and novel miR-94 showed a falling then rising trend, which was in line with the expression trend in the sequencing data, showing that the small RNA sequencing results were accurate (Figure 6).

## 4. Discussion

Antler growth is primarily driven and controlled by the antler apical growth center—which is made up of velvet skin, mesenchyme, and cartilage tissue—by the process of endochondral ossification [1,9]. The antler skin and cartilage tissues in the antler growth center are rich in vascular tissue and differ greatly from normal skin and cartilage tissues, which is inevitably associated with the astonishing growth rate of the antler and its capacity for repair and regeneration [29]. Therefore, the investigation of changes in gene expression in antler growth centers during different growth periods is highly significant in order to reveal the mechanism of rapid antler growth. The majority of organ growth is driven by the combinatorial effects of cell proliferation and cell expansion [30].

The purpose of this study was to deep sequence for miRNAs in velvet antler growth centers at different growth stages. In total, 292, 102, and 60 miRNAs were found to be differentially expressed in the d30 vs. d60, d30 vs. d90, and d60 vs. d90 comparison groups, respectively. Furthermore, 10 miRNAs were differentially expressed in the d30 vs. d60 and d60 vs. d90 comparison groups but not in the d30 vs. d90 group, among which four differentially expressed miRNAs (ppy-miR-1, mmu-miR-200b-3p, novel miR-6, and novel miR-94) with high expression levels in the antler growth center of 30 and 90 days and low expression in the antler growth center of 60 days. Meanwhile, the reverse was true for another miRNA (novel miR-10). Given that miRNAs typically function by down-regulating the expression of their target genes [31], five differentiated miRNAs were screened, which could play important roles in the rapid growth of velvet antlers. Further validation is required to determine whether miRNAs that are highly or lowly expressed in antler growth centers at 60 days of growth promote antler cell proliferation by suppressing the expression of their target genes.

It has been reported that miR-1 has a broad range of biological functions, e.g., in modulating skeletal muscle proliferation and differentiation [32,33], playing roles in heart disease [34,35,36,37], sex determination [38], and diabetes [39]. MiR-1 also plays critical roles in chondrogenesis [40]; the proliferation, differentiation, and hypertrophy of chondrocytes [41,42]; and the development of osteoarthritis [43,44]. The growth center of the antler tip drives the longitudinal endochondral ossification process of velvet antlers [45,46]. It is probable that miR-1 has a significant impact on this procedure. In one study, it was discovered that miR-1 was capable of suppressing the proliferation of chondrocytes in Chinese sika deer by specifically targeting the 3′UTR of IGF-1 [47]. In the present study, ppy-miR-1 was lowly expressed in the growth center of antlers at 60 days of growth and highly expressed at 30 and 90 days of growth, which may be closely related to the rapid growth and ossification of antlers.

MiR-200b is a miRNA with a wide range of functions, including functions in multiple cancers or tumors [48,49,50,51], endotheliogenesis [52,53,54], angiogenesis [55], etc. It has been reported that a decreased level of miR-200b-3p may result in angiogenesis in HCC [56]. The growing tips of velvet antlers are rich in blood vessels [4,19,57,58,59]. The antler grows quickly, including its interior components, such as the blood vessels, nerves, and skin that join it [59]. Mmu-miR-200b-3p may promote angiogenesis in antlers during the rapid growth phase, which would provide nutritional and other types of support for the rapid growth of antlers.

The novel miR-94, the target genes of which are strongly annotated to the “Pathways in cancer” and “Wnt signaling pathway”, is the miRNA in which we are most interested. It was shown that the proliferation of velvet antler cells was faster than that of cancer cells during the rapid growth stage of antlers [23,60,61]. Academics refer to this phenomenon as cancer-like growth [62,63]. “Pathways in cancer” is a complicated signaling pathway, forming a complex network with the “ERK signaling pathway”, “PI3K-AKT signaling pathway”, “Wnt signaling pathway”, “Notch signaling pathway”, “TGF-β signaling pathway”, “HIF-1 signaling pathway”, and other signaling pathways. Studies showed that the “ERK signaling pathway” [64], “PI3K-AKT signaling pathway” [65,66,67], “Notch signaling pathway” [68,69], “TGF-β signaling pathway” [70,71], “Wnt signaling pathway” [72,73,74], and “HIF-1 signaling pathway” [66] were involved in the development and regeneration of velvet antlers. Thus, more research is necessary to determine if novel miR-94 and its target genes have a major impact on the quick development and even the regeneration of velvet antlers.

The target genes of novel miR-6 annotated fewer signaling pathways than ppy-miR-1, mmu-miR-200b-3p, and novel miR-94, and the target genes are significantly enriched in “mTOR signaling pathway”, “Pathways in cancer”, “Rap1 signaling pathway” and “signaling pathways regulating pluripotency of stem cells”. mTOR is a vital regulator of multiple cellular functions, including cell proliferation, differentiation, and protein synthesis [75]. The mTOR signaling pathway can stimulate the protein synthesis of chondrocytes to regulate mammalian limb skeletal growth [76]. Studies have indicated that mTOR is an important regulator of cartilage development [77,78,79,80]. The Rap1 signaling pathway is a cancer-related signaling pathway with a wide range of biological functions [81,82,83,84]. The longitudinal section of the antler was divided into velvet skin, mesenchyme, pre-cartilage, and cartilage from top to bottom, which clearly reflected the dynamic process of antler ossification, indicating that the growth and development of antler is essentially the process of bone tissue growth and development [85]. All factors that theoretically play a relevant role in bone tissue development may be involved in the regulation of antler growth and development. Studies on the regulation of antler growth by the mTOR signaling pathway and Rap1 signaling pathway have not been reported yet. Whether novel miR-6 regulates the proliferation of antler cells through these two pathways remains to be further verified. The antler mesenchyme is filled with mesenchymal cells (reserve mesenchyme cells, RMCs) that have the ability to divide vigorously and differentiate towards chondrocytes, which ultimately secrete the bone matrix to complete the ossification [7]. RMCs are stem cells that drive the rapid growth of velvet antlers and are stromal cells capable of self-replication and have the potential to differentiate into chondrocytes, etc. [86]. Target genes of novel miR-6 annotated to the “Signaling pathways regulating pluripotency of stem cells”, this miRNA and its target genes may be involved in the differentiation of RMCs into chondrocytes. Taken together, the screened DEMs have a certain significance for the study of the rapid growth of velvet antlers, and the regulatory mechanism remains to be further investigated.

## 5. Conclusions

In summary, we detected miRNAs that were differentially expressed in antler growth centers at different stages of growth using an RNA-seq approach. Following the comparison and analysis, we screened five differentially expressed miRNAs with target genes that were significantly enriched in the “Wnt signaling pathway”, “PI3K-Akt signaling pathway”, “MAPK signaling pathway”, “TGF-β signaling pathway”, “mTOR signaling pathway”, and other classical signaling pathways. These miRNAs, especially ppy-miR-1, mmu-miR-200b-3p, and novel miR-94, could be extremely important for the rapid development of antlers.

## Figures and Tables

**Figure 1 genes-14-00424-f001:**
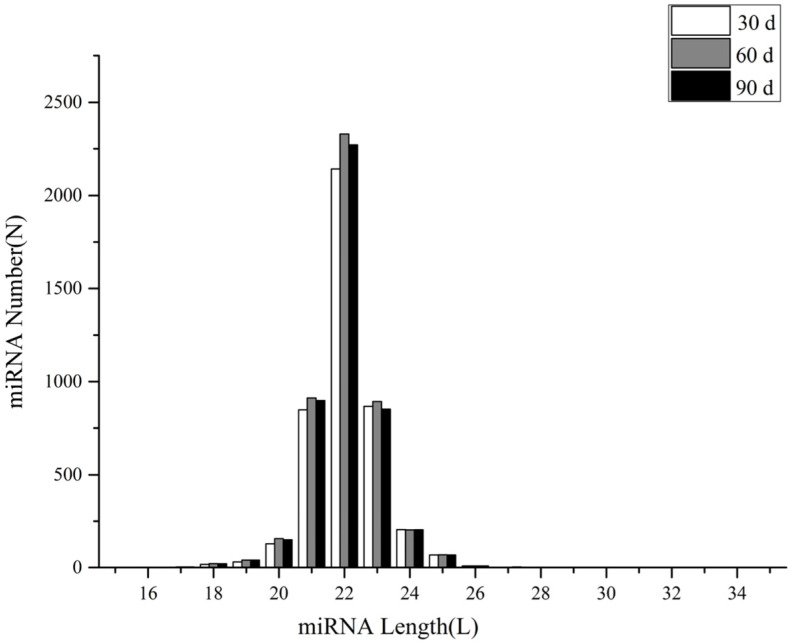
Length distribution of all identified miRNAs. The sequence length of mature miRNA in most species is about 22 nt. The length distribution of miRNAs identified in this study was 18–27 nt, mainly ranging from 20 to 24 nt.

**Figure 2 genes-14-00424-f002:**
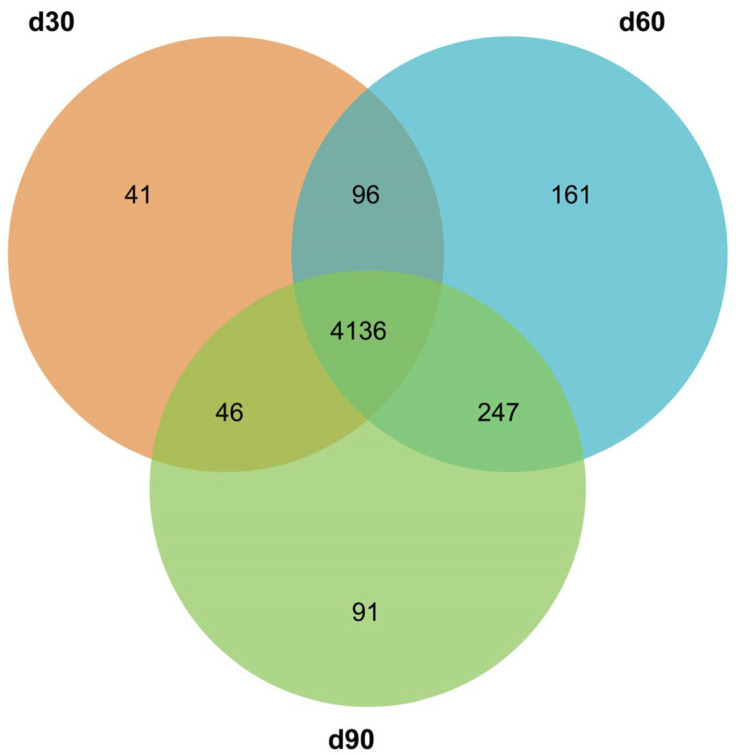
Venn diagram of all miRNAs. It has identified 4818 miRNAs in antler growth centers of *Cervus elaphus kansuensis* at three development stages (30 d, 60 d, 90 d). Most of the miRNAs (4136) were co-expressed in three libraries. Only 41, 161, and 91 miRNAs were specially expressed in d30, d60, and d90 antler growth centers.

**Figure 3 genes-14-00424-f003:**
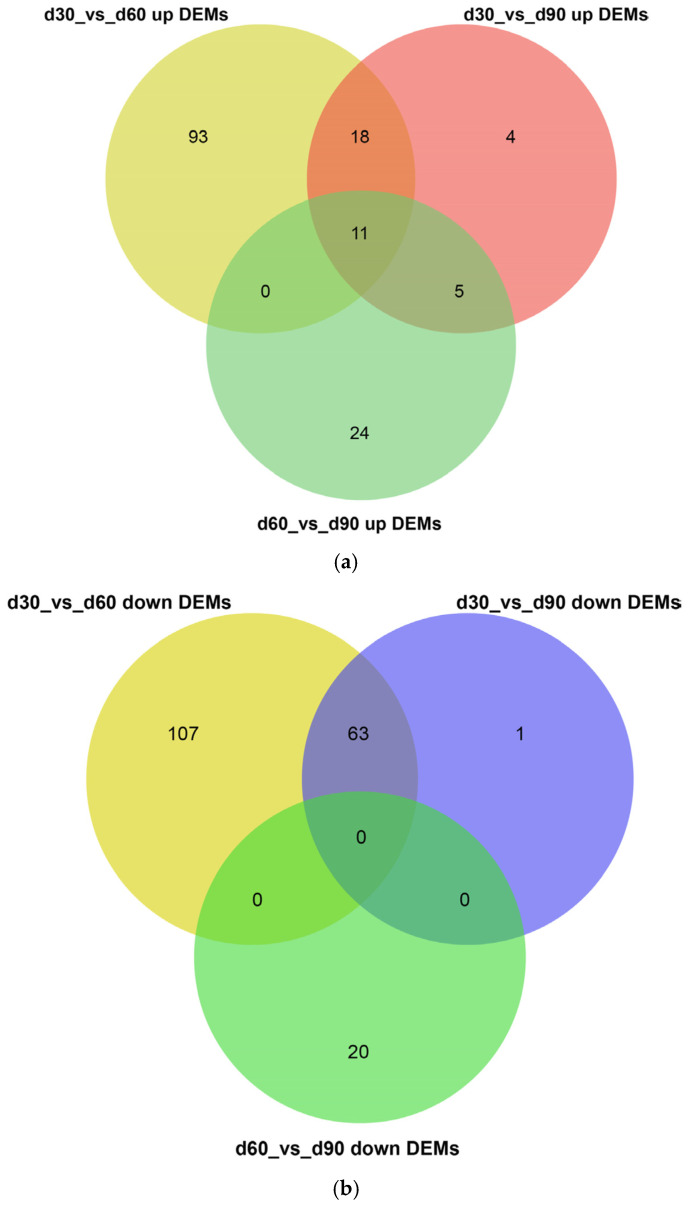
Venn diagram of differentially expressed miRNAs. In the Venn diagram, the number of overlapping triangular regions represents the number of common DEMs among the corresponding comparison combinations (d30 vs. d60, d90 vs. d60, and d30 vs. d90), and the non-overlapping region represents the unique differential genes in each comparison combination. (**a**) The up-regulated DEMs in three comparison groups; (**b**) The down-regulated DEMs in three comparison groups.

**Figure 4 genes-14-00424-f004:**
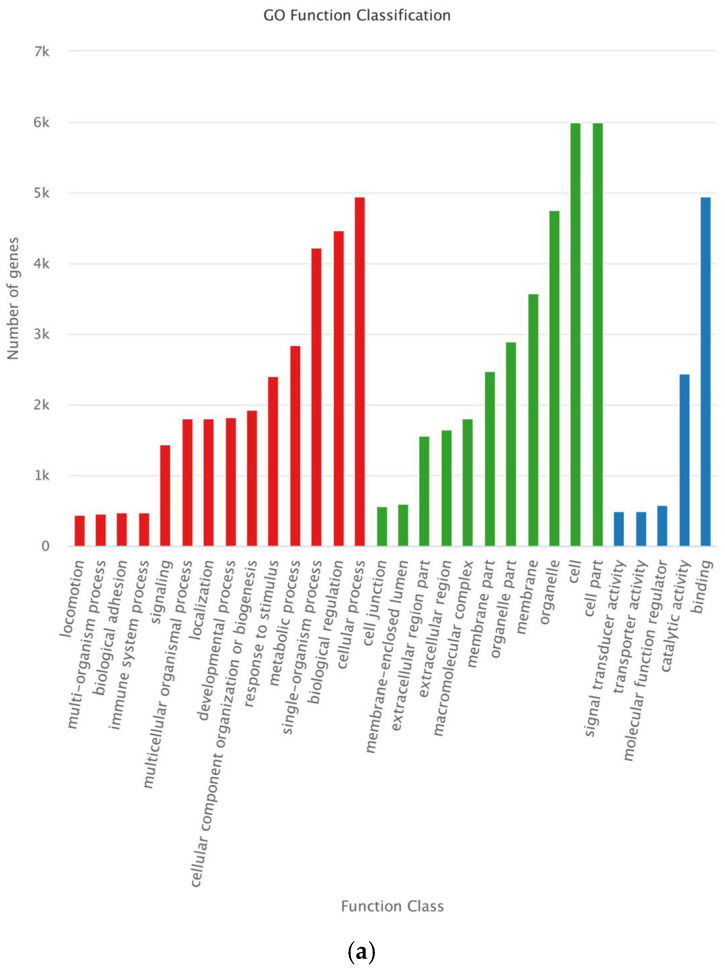
GO classification of target genes for DEMs in different comparison groups. (**a**) GO classification of target genes for DEMs in d30 vs. d60; (**b**) GO classification of target genes for DEMs in d30 vs. d90; (**c**) GO classification of target genes for DEMs in d60 vs. d90.

**Figure 5 genes-14-00424-f005:**
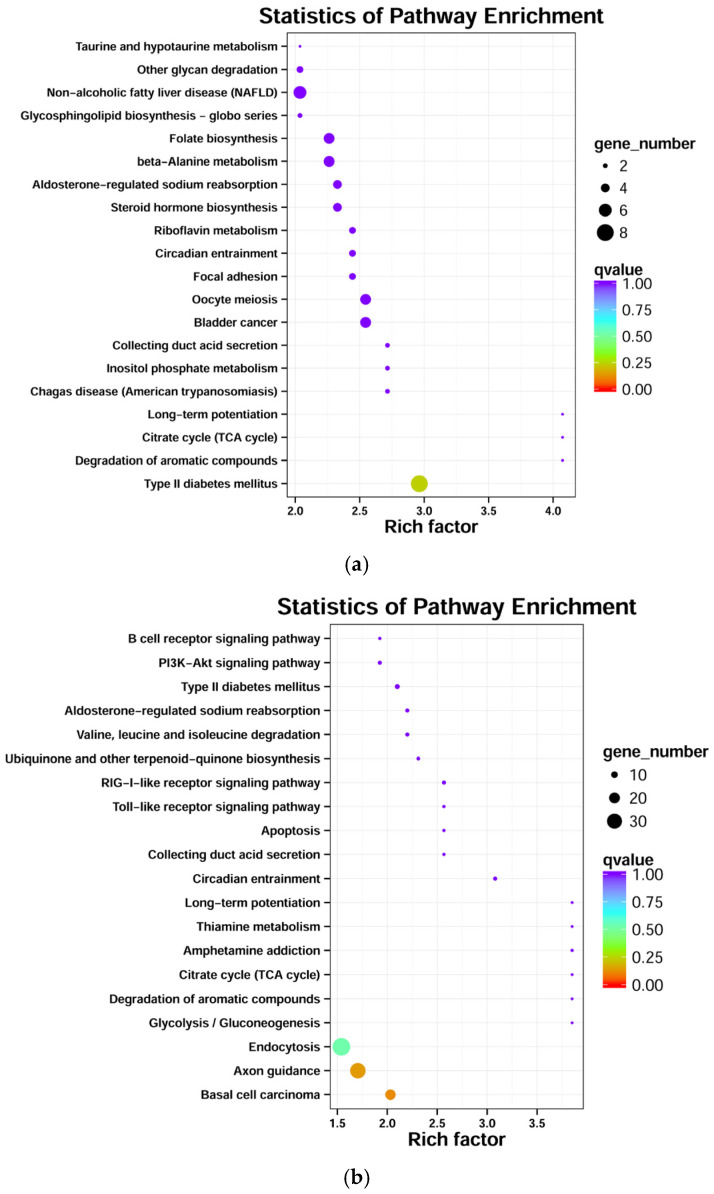
KEGG pathway enrichment of target genes for DEMs in different comparison groups. (**a**) KEGG pathway enrichment of target genes for DEMs in d30 vs. d60; (**b**) KEGG pathway enrichment of target genes for DEMs in d30 vs. d90; (**c**) KEGG pathway enrichment of target genes for DEMs in d60 vs. d90.

**Figure 6 genes-14-00424-f006:**
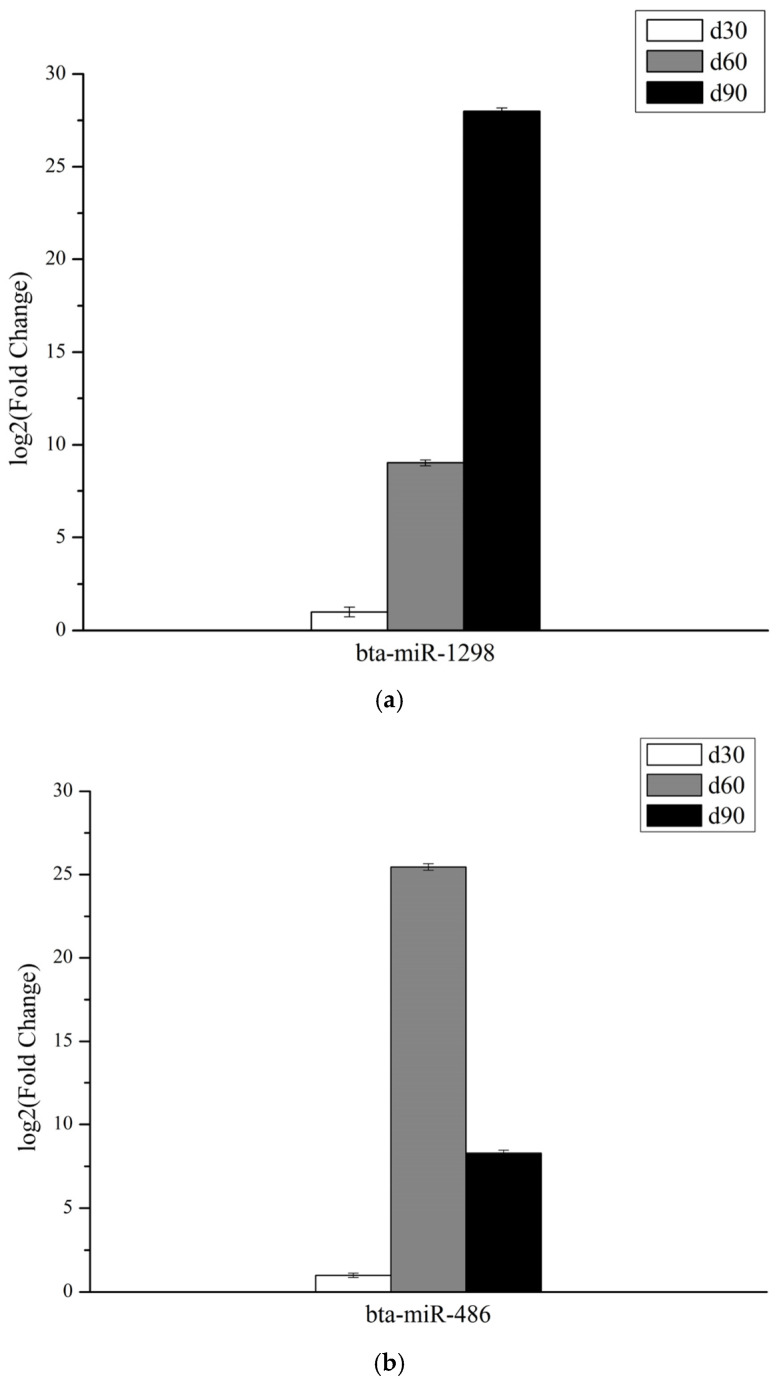
Results of qPCR validation. MicroRNA expression was assessed by qPCR in growth centers of velvet antlers in three growth stages. The relative expression levels were calculated by the 2^−ΔΔCt^ method. (**a**) bta−miR−1298; (**b**) bta−miR−486; (**c**) novel miR−166; (**d**) ppy−miR−1; (**e**) mmu−miR−200b−3p; (**f**) novel miR−6; (**g**) novel miR−94.

**Table 1 genes-14-00424-t001:** Primers used for reverse transcription and stem-loop real-time qPCR.

Gene ID	miRNA Sequence	Primers	Sequence
*U6*		RTP	AACGCTTCACGAATTTGCGT
		Forward primer	CTCGCTTCGGCAGCACA
		Reverse primer	AACGCTTCACGAATTTGCG
bta-miR-1298	UUCAUUCGGCUGUCCAGAUGUA	RTP	CTCAACTGGTGTCGTGGAGTCGGCAATTCAGTTGAG TACATCTG
		Forward primer	ACACTCCAGCTGGGUUCAUUCGGCUGUCCA
		Reverse primer	TGTCGTGGAGTCGGCAATTCAG
bta-miR-486	UCCUGUACUGAGCUGCCCCGAG	RTP	CTCAACTGGTGTCGTGGAGTCGGCAATTCAGTTGAG CTCGGGGC
		Forward primer	ACACTCCAGCTGGGUCCUGUACUGAGCUGC
		Reverse primer	TGTCGTGGAGTCGGCAATTCAG
novel miR-166	GUGGGCUUCCCUGGUGGCUCAGA	RTP	CTCAACTGGTGTCGTGGAGTCGGCAATTCAGTTGAG TCTGAGCC
		Forward primer	ACACTCCAGCTGGGGUGGGCUUCCCUGGUGG
		Reverse primer	TGTCGTGGAGTCGGCAATTCAG
ppy-miR-1	UGGAAUGUAAAGAAGUAUGUAU	RTP	CTCAACTGGTGTCGTGGAGTCGGCAATTCAGTTGAG ATACATAC
		Forward primer	ACACTCCAGCTGGG UGGAAUGUAAAGAAGU
		Reverse primer	TGTCGTGGAGTCGGCAATTCAG
mmu-miR-200b-3p	UAAUACUGCCUGGUAAUGAUGA	RTP	CTCAACTGGTGTCGTGGAGTCGGCAATTCAGTTGAG TCATCATT
		Forward primer	ACACTCCAGCTGGGUAAUACUGCCUGGUAA
		Reverse primer	TGTCGTGGAGTCGGCAATTCAG
Novel miR-6	CAAAUUCGUGAAGCGUUCCAUAUUU	RTP	CTCAACTGGTGTCGTGGAGTCGGCAATTCAGTTGAG AAATATGG
		Forward primer	ACACTCCAGCTGGGCAAAUUCGUGAAGCGUUCC
		Reverse primer	TGTCGTGGAGTCGGCAATTCAG
Novel miR-94	GAAUUAUAGGAAUUGAACC	RTP	CTCAACTGGTGTCGTGGAGTCGGCAATTCAGTTGAG GGTTCAAT
		Forward primer	ACACTCCAGCTGGGGAAUUAUAGGAAU
		Reverse primer	TGTCGTGGAGTCGGCAATTCAG

**Table 2 genes-14-00424-t002:** The top 20 known miRNAs with the highest expression abundance in antler growth centers at different growth stages.

	d30			d60			d90	
miRNA ID	Count	TPM	miRNA ID	Count	TPM	miRNA ID	Count	TPM
xla-miR-148a-3p	141,308	22,996.78	xla-miR-148a-3p	154,290	17,939.06	xla-miR-148a-3p	143,637	21,242.91
chi-miR-148a-3p	131,464	21,394.74	chi-miR-148a-3p	144,249	16,771.61	chi-miR-148a-3p	131,088	19,387
mmu-miR-148a-3p	131,464	21,394.74	mmu-miR-148a-3p	144,249	16,771.61	mmu-miR-148a-3p	131,088	19,387
ggo-miR-148a	131,461	21,394.25	ggo-miR-148a	144,248	16,771.5	ggo-miR-148a	131,087	19,386.85
rno-miR-148a-3p	131,461	21,394.25	rno-miR-148a-3p	144,248	16,771.5	rno-miR-148a-3p	131,087	19,386.85
oan-miR-148-3p	131,423	21,388.07	oan-miR-148-3p	144,240	16,770.57	oan-miR-148-3p	131,062	19,383.16
bta-miR-148a	131,378	21,380.75	bta-miR-148a	144,213	16,767.43	bta-miR-148a	131,046	19,380.79
cgr-miR-148a	131,378	21,380.75	cgr-miR-148a	144,213	16,767.43	cgr-miR-148a	131,046	19,380.79
eca-miR-148a	131,378	21,380.75	eca-miR-148a	144,213	16,767.43	eca-miR-148a	131,046	19,380.79
hsa-miR-148a-3p	131,378	21,380.75	hsa-miR-148a-3p	144,213	16,767.43	hsa-miR-148a-3p	131,046	19,380.79
mdo-miR-148-3p	131,378	21,380.75	mdo-miR-148-3p	144,213	16,767.43	mdo-miR-148-3p	131,046	19,380.79
mml-miR-148a-3p	131,378	21,380.75	mml-miR-148a-3p	144,213	16,767.43	mml-miR-148a-3p	131,046	19,380.79
oar-miR-148a	131,378	21,380.75	oar-miR-148a	144,213	16,767.43	oar-miR-148a	131,046	19,380.79
ppy-miR-148a	131,378	21,380.75	ppy-miR-148a	144,213	16,767.43	ppy-miR-148a	131,046	19,380.79
ssc-miR-148a-3p	131,378	21,380.75	ssc-miR-148a-3p	144,213	16,767.43	ssc-miR-148a-3p	131,046	19,380.79
ptr-miR-148a	131,336	21,373.91	ptr-miR-148a	144,189	16,764.64	ptr-miR-148a	131,017	19,376.5
ppa-miR-148a	130,305	21,206.12	ppa-miR-148a	143,467	16,680.69	ppa-miR-148a	130,182	19,253.01
cfa-miR-148a	128,867	20,972.1	cfa-miR-148a	142,161	16,528.84	cfa-miR-148a	128,908	19,064.59
ocu-miR-148a-3p	128,867	20,972.1	ocu-miR-148a-3p	142,161	16,528.84	ocu-miR-148a-3p	128,908	19,064.59
tch-miR-148a-3p	128,867	20,972.1	tch-miR-148a-3p	142,161	16,528.84	tch-miR-148a-3p	128,908	19,064.59

**Table 3 genes-14-00424-t003:** The top 20 novel miRNAs with the highest expression abundance in antler growth centers at different growth stages.

	d30			d60			d90	
miRNA ID	Count	TPM	miRNA ID	Count	TPM	miRNA ID	Count	TPM
novel miR-249	683	111.1529	novel miR-249	1171	136.1504	novel miR-249	1008	149.0762
novel miR-77	654	106.4334	novel miR-77	782	90.92195	novel miR-77	684	101.1588
novel miR-93	274	44.59137	novel miR-93	521	60.57588	novel miR-166	457	67.58711
novel miR-15	187	30.4328	novel miR-15	311	36.1595	novel miR-267	456	67.43922
novel miR-114	177	28.80537	novel miR-114	207	24.06758	novel miR-5	456	67.43922
novel miR-6	138	22.45843	novel miR-57	164	19.06803	novel miR-86	456	67.43922
novel miR-150	121	19.69181	novel miR-21	118	13.71968	novel miR-93	413	61.07982
novel miR-57	102	16.59971	novel miR-150	115	13.37088	novel miR-52	290	42.88898
novel miR-91	102	16.59971	novel miR-44	115	13.37088	novel miR-15	226	33.42382
novel miR-48	101	16.43696	novel miR-10	112	13.02207	novel miR-114	183	27.06442
novel miR-84	94	15.29777	novel miR-91	108	12.557	novel miR-57	142	21.00081
novel miR-83	91	14.80954	novel miR-48	102	11.85939	novel miR-94	118	17.45138
novel miR-72	64	10.4155	novel miR-265	98	11.39431	novel miR-48	114	16.85981
novel miR-193	62	10.09002	novel miR-84	98	11.39431	novel miR-6	114	16.85981
novel miR-21	61	9.927275	novel miR-247	82	9.534016	novel miR-150	109	16.12034
novel miR-94	60	9.764533	novel miR-41	77	8.952673	novel miR-265	100	14.7893
novel miR-71	57	9.276307	novel miR-107	76	8.836405	novel miR-91	93	13.75405
novel miR-107	55	8.950822	novel miR-83	76	8.836405	novel miR-81	82	12.12723
novel miR-41	55	8.950822	novel miR-71	68	7.906257	novel miR-44	72	10.6483
novel miR-90	49	7.974369	novel miR-6	64	7.441183	novel miR-72	72	10.6483

## Data Availability

Raw data were submitted to the National Center for Biotechnology Information (NCBI) Sequence Read Archive (SRA) database (registry number SUB12298775, BioProject: PRJNA903551). Available online: https://submit.ncbi.nlm.nih.gov/subs/bioproject/SUB12298775/overview.

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
