# Peer review of "MiRNA Profiling and Its Potential Roles in Rapid Growth of Velvet Antler in Gansu Red Deer (Cervus elaphus kansuensis)"

_genes, 2023, doi:10.3390/genes14020424_

Round 1

Reviewer 1 Report

The topic of this manuscript is very interesting, in which the authors conducted miRNA sequencing at 3 phases of rapid antler growth from the antler growth centers.  This type of research is useful, and with sufficient experimental findings, may shed light on the mechanisms that drive antler growth and development.  However, the manuscript fails to demonstrate insightful or thoughtful conclusions from the generated sequence data and subsequent analyses.    While the manuscript is largely grammatically adequate, the writing implies a lack of understanding of certain physiological processes. 

The experimental design is essentially a pilot study and uses only 3 animals at 3 time points.  The sample choice increases the complexity of the study.  These data should be used to design a carefully planned experiment that would enable more insight into antler growth.

Specific comments:

The abstract contains an excess of introductory material that could be moved to the introduction.  The conclusions are weak and not supported by  evidence -- differentially expressed microRNAs were indeed identified and may play a role in antler development, but no experimental evidence other than sequence is presented.

Line 30. The first sentence should be revised - antlers in the velvet stage of growth are cartilaginous protrusions from the frontal bone of male deer.  They are not horns.

What was the overall antler length at each stage?  What percentage of the total length was the apical 5cm that was harvested? 

Were the same 3 animals used for all 3 time points?  Does injury/repair affect results?

Line 36.  ...growth is organized and not neoplastic. The antlers eventually undergo complete ossification.

Line 38-39  Comparison to malignant growth here is not really appropriate, or at least the meaning is not correctly described.  It's fair to contrast with neoplasia - which is dysregulated growth that can be benign, or described as malignant when the growth invades other tissues.  Antler growth is much different than malignant growth.

Line 41.  Many portions of the ms should be edited for clarity.  For example, "the growing season [growth rate (?)] of antlers varies from slow spring [growth] to exponentially accelerated [growth] in summer that slows in autumn...

Line 86 should specify 3 stages of [antler] growth for clarity

Line 87-88 If the dermis, mesenchyme, and cartilage were separated, why was RNA extraction and sequence not conducted separately on the three tissue types.  Was epidermis and vasculature also included in tissue for RNA extraction?

Line 92. Were the anesthesia and surgical procedures conducted by or supervised by a veterinarian?  What animal approval was granted for this study?

Line106-134.  What parameters used for evaluating library quality? Sequence output, read depth, quality, seems to be missing in this section, but can be found in the results.  Perhaps include some detail here about assessment methods here. 

Very interesting that so many novel miRNAs were identified.  Particularly given this novelty, the Targetscan analysis is not particularly useful since it detects a high percentage of false positives.   Rather than generic categories of cellular processes, the manuscript would benefit from evaluation of gene networks with respect to the biology of antler growth.  Gene and signalling pathways are tissue-specific fairly well described in other mammals for chondrogenesis, as well as epithelial, dermal, and vascular development.  Thus ras, PI3K-Akt, Mapk, mTOR, and TGF-beta are not surprising.  While these are often altered in cancer and identified in carcinogenesis studies, they are also active in epithelial growth, for example.

Line 286 Figure 6 Please show error bars.  With only 3 samples, were the results consistent across the 3 animals for each time point and microRNA?

For each miRNA, is the day30 sample expression set at 1 for relative expression calculation, and each miRNA is evaluated separately?  If yes, this would be appropriate but graphing the miR expression for all miRs within the same graph is misleading since gene to gene expression is not directly comparable.  Better to plot these separately

Would be very interesting to relate this gene expression to histology of the growth stage.

Lines 290-293.  The first two sentences could be deleted.

Lines 317-327.  How do you explain the low expression of several of the miRNAs at d60 compared to d30 and d90?

Line 356-358.  These miRNAs have been shown to be differentially expressed at 3 stages of antler development.  They could be important, yes, but aside from demonstrating differential expression, this research offers no evidence of how they might participate, and does not describe where/which tissue type the expression occurs. 

This study is a very interesting first step.  The authors are encouraged to spend more time on the implications of these results and clarify the questions asked.

Appreciate that the data are submitted to a public data base.  Good luck with future efforts.

Author Response

Response to Reviewer 1

Comments and Suggestions for Authors

Q: The topic of this manuscript is very interesting, in which the authors conducted miRNA sequencing at 3 phases of rapid antler growth from the antler growth centers. This type of research is useful, and with sufficient experimental findings, may shed light on the mechanisms that drive antler growth and development. However, the manuscript fails to demonstrate insightful or thoughtful conclusions from the generated sequence data and subsequent analyses. While the manuscript is largely grammatically adequate, the writing implies a lack of understanding of certain physiological processes. 

A: Many thanks to the reviewer for your comments on our manuscript. In this study, analysis of high-throughput sequencing data revealed that ppy-miR-1, mmu-miR-200b-3p, and novel miR-94 may play crucial roles in the rapid antler growth in summer. Subsequently, we will validate the functions of these miRNAs at the cellular level. In addition, our manuscript has undergone English language editing by MDPI. We hope that our manuscript will be free of grammatical errors and that our study will be clearly presented.

Q: The experimental design is essentially a pilot study and uses only 3 animals at 3 time points. The sample choice increases the complexity of the study. These data should be used to design a carefully planned experiment that would enable more insight into antler growth.

A: In this study, 3 male red deer were selected to collect antler samples at about 30 d, 60 d and 90 d growth stages, so that there were 3 samples from different individuals at each time point. In high-throughput sequencing, each sample pool was a mix of equal amount of RNAs from three individual male deer, as was the case in other studies (Mundalil Vasu M, et al. 2014; Jia BY, et al. 2016; Chen CF, et al. 2019; Hu P, et al. 2019; Jia B, et al. 2020). We are sorry that this manuscript uses RNA pools for RNA-Seq instead of three biological replicates, but this method is acceptable (Jia B, et al. 2021).

Specific comments:

Q: The abstract contains an excess of introductory material that could be moved to the introduction. The conclusions are weak and not supported by evidence -- differentially expressed microRNAs were indeed identified and may play a role in antler development, but no experimental evidence other than sequence is presented.

A: The abstract has been modified. This manuscript is a pure RNA-seq analysis study, from which we screened sevel miRNAs that may play biological functions in the rapid growth of velvet antler, and detected the expression of these miRNAs in antler growth centers at different growth periods by qPCR. We are currently culturing velvet antler cells and will further validate the functions and molecular mechanisms of these miRNAs in antler cell proliferation later, and the results will be published in the future.

Q: Line 30. The first sentence should be revised - antlers in the velvet stage of growth are cartilaginous protrusions from the frontal bone of male deer. They are not horns.

A: That sentence has be revised (Line 35-36)

Q: What was the overall antler length at each stage? What percentage of the total length was the apical 5cm that was harvested? 

A: We are sorry that we did not have statistics on the above data.

Q: Were the same 3 animals used for all 3 time points? Does injury/repair affect results?

A: To avoid inaccurate results caused by individual differences, we collected antler samples from the same 3 animals at 3 time points. Each deer had a pair of antlers, so we removed one of them when they were about 30 days of growth, followed by the lateral branch of the other antler when they were about 60 days of growth, and finally the main branch of the remaining antler when they were about 90 days of growth.

Q: Line 36.  ...growth is organized and not neoplastic. The antlers eventually undergo complete ossification.

A: Thank you very much for your comments. We have revised it in the manuscript (Line 43-44).

Q: Line 38-39  Comparison to malignant growth here is not really appropriate, or at least the meaning is not correctly described.  It's fair to contrast with neoplasia - which is dysregulated growth that can be benign, or described as malignant when the growth invades other tissues.  Antler growth is much different than malignant growth.

A: That statement was just to explain that deer antlers proliferate much faster than that of cancer cells, and that they can stop proliferating in time, unlike the continuous proliferation of cancer cells. We are not sure how that sentence should be revised, so please give us your more detailed comments, thank you!

Q: Line 41.  Many portions of the ms should be edited for clarity. For example, "the growing season [growth rate (?)] of antlers varies from slow spring [growth] to exponentially accelerated [growth] in summer that slows in autumn...

A: Scholars have only studied the growth rate of antlers during the rapid growth period, which is about 2 cm/day. We are sorry that we have not yet seen any reports on the growth rate of antlers during the early growth and ossification periods.

Q: Line 86 should specify 3 stages of [antler] growth for clarity

A: The growth characteristics of velvet antler have been described in the “Introduction” section (Line 52-58).

Q: Line 87-88 If the dermis, mesenchyme, and cartilage were separated, why was RNA extraction and sequence not conducted separately on the three tissue types. Was epidermis and vasculature also included in tissue for RNA extraction?

A: The rapid growth of velvet antler is driven by the growth center. In this study, the growth centers of velvet antler were taken as a whole to detect the differences in miRNA expression in the growth centers of velvet antler at different growth stages, in order to provide a basis for the research on the rapid growth of velvet antler. This is similar to other reports (Yao B, et al. 2019; Jia B, et al. 2021). The growth centers of an antler mainly include the antler skin, mesenchyme, and cartilage tissue and antler is rich in blood vessels, therefore, the RNA extraction in this study included deer antler dermal tissue and blood vessels.

Q: Line 92. Were the anesthesia and surgical procedures conducted by or supervised by a veterinarian?  What animal approval was granted for this study?

A: In this study, all experimental protocols were approved by the Institutional Animal Care and Use Committee of Qinghai University (Xining, China), and all methods were carried out to approved guidelines and regulations (Code: SL-2022024). “Ethics approval” was added after the “Conclusion” section. The anesthesia and surgical procedures were conducted by a professional veterinarian.

Q: Line106-134. What parameters used for evaluating library quality? Sequence output, read depth, quality, seems to be missing in this section, but can be found in the results.  Perhaps include some detail here about assessment methods here. 

A: Both data processing and functional annotation of miRNA target genes were processed according to the default parameters of the software and database. Part of them has been supplemented in the manuscript (Line 158).

Q: Very interesting that so many novel miRNAs were identified. Particularly given this novelty, the Targetscan analysis is not particularly useful since it detects a high percentage of false positives.  Rather than generic categories of cellular processes, the manuscript would benefit from evaluation of gene networks with respect to the biology of antler growth. Gene and signalling pathways are tissue-specific fairly well described in other mammals for chondrogenesis, as well as epithelial, dermal, and vascular development. Thus ras, PI3K-Akt, Mapk, mTOR, and TGF-beta are not surprising. While these are often altered in cancer and identified in carcinogenesis studies, they are also active in epithelial growth, for example.

A: In this study, miRNA seed regions were used to predict the target genes of new mirnas through Targetscan database, and the function of target genes of miRNAs was annotated by KEGG database. They were only predictions of target genes and gene functions, and the real function of these genes needs to be verified later. Ras, PI3K-Akt, Mapk, mTOR, and TGF-beta are all classical signaling pathways with a wide range of biological functions. The miRNAs and their target genes annotated to these signaling pathways are the focus of our subsequent studies

Q: Line 286 Figure 6 Please show error bars.  With only 3 samples, were the results consistent across the 3 animals for each time point and microRNA?

A: Figure 6 added the error lines and the figure was replaced. In this study, three antler samples were collected at each time point. In the qPCR experiment, 9 samples at 3 time points were tested, and three replicates were performed for each sample. The average of the nine Ct values at each time point was used to compute the relative expression of each miRNA.

Q: For each miRNA, is the day30 sample expression set at 1 for relative expression calculation, and each miRNA is evaluated separately?  If yes, this would be appropriate but graphing the miR expression for all miRs within the same graph is misleading since gene to gene expression is not directly comparable. Better to plot these separately

A: Relative gene expression was subsequently analyzed using the comparative 2ΔΔCt method (Livak & Schmittgen, 2001; Rao et al., 2013), and each miRNA was evaluated separately. The relative quantitative qPCR approach was applied in this study. Similar to other findings, the relative expression of a single miRNA could be displayed in its own separate figure or the relative expression of multiple miRNAs could be displayed in a single figure (Liu G, 2022; Zhang QL, 2019; Chu W, 2017; He R, 2017).

Q: Would be very interesting to relate this gene expression to histology of the growth stage.

A: The differentially expressed miRNAs were selected for qPCR only to verify the accuracy of high-throughput sequencing data, so the expression of screened miRNAs in antler growth centers at each growth stage was detected. This is similar to other reports (Wu P, et al. 2021; Wang C, et al. 2022).

Q: Lines 290-293. The first two sentences could be deleted.

A: These two sentences have been deleted (Line 330-333).

Q: Lines 317-327. How do you explain the low expression of several of the miRNAs at d60 compared to d30 and d90?

A: MiRNAs usually negatively regulate the expression of target genes at the transcriptional or post-transcriptional level. For ppy-miR-1, it has been shown that miR-1 was specifically targeted the 3'UTR of IGF-1, which is highly expressed in antler during the rapid growth period (Sadighi M, et al. 2001; Yang F, et al. 2014). In this study, ppy-miR-1 was found to have low expression in the antler growth center at 60 days of growth and high expression at 30 and 90 days of antlers. This may indicate that miRNA-1 had low expression during the period of rapid antler growth, which allowed the negative regulatory influence on IGF-1 to be released and encouraged the growth of velvet antler.

Q: Line 356-358. These miRNAs have been shown to be differentially expressed at 3 stages of antler development. They could be important, yes, but aside from demonstrating differential expression, this research offers no evidence of how they might participate, and does not describe where/which tissue type the expression occurs. 

A: In this study, analysis of high-throughput sequencing data revealed that ppy-miR-1, mmu-miR-200b-3p, and novel miR-94 might play crucial roles in the rapid antler growth in summer. These miRNAs were screened based on functional annotations of their target genes. Subsequently, we will validate the functions of these miRNAs at the cellular level.

This study is a very interesting first step. The authors are encouraged to spend more time on the implications of these results and clarify the questions asked.

Appreciate that the data are submitted to a public data base. Good luck with future efforts.

Thank you very much for the reviewers' comments and encouragement on our manuscript! We have revised the manuscript and hope to make it more rigorous and scientific and will be accepted by the journal.

Reviewer 2 Report

The research topic is relevant and interesting, but some limitations were found that could be addressed. I would also suggest that the manuscript should meet a professional English editing company.  

Abstract:

The research theme is not clear in the abstract; What are the market value and opportunities for velvet antlers and why do the authors design this study? Please highlight the clear and concise theme in the abstract.

Line; 19;  ( In an effort), modify the term.

Line; 24; (which were associated to the growth and development of velvet antlers); what is the difference between growth and development? Please revise the sentence.

 Introduction:

The introduction is confusing and lacking in concepts. For example: Line 31-32; It is the only mammalian organ capable of complete regeneration. 

Line; 38-39; The research theme is not appropriate, revise it.

Lines; 44 and 49; (in winter; which is) the initials are not correct, grammatically, so revise it.  

Lines; 58-59; rephrased the sentence and remove the term scholars.

Line; 65; (in a totally or partially); what does it mean? I would suggest, correcting the sentence and rephrased.

Material and methods:

2.1. Sample Collection

I did not see any ethical statement here. How do they care for the experimental animals, were any necessary measures taken to keep the experimental animals as pain-free as possible? What was the most effective and reliable anesthesia for the removal of antlers and how do administer the anesthesia?

Line; 98; how do the contaminants are removed from the extracted RNS?

Lines; 118-119; The sentence should be excluded from the method section.

Lines; 122-123; (for sequences that could not be aligned to the miR-123 Base database but could be mapped to the red deer genome); what does it mean?

Line; 128; (we compared DEMs in a two-way comparison 128 between d30 versus d60, d30 versus d90 and d60 versus d90), Is it a two-way comparison? Please elaborate..

Results:

Figure 3. Venn diagram of differentially expressed miRNAs. Please explain the legends comprehensively.

I did not see the Hierarchical diagram of differentially expressed miRNAs.  

Figure 6: how can we compare the validation results to the sequencing results, because the manuscript does not have the graph for Seq data e.g., (seq data vs RTqPCR)…

Discussion:

The discussion is too short and repetition of information found in the introduction and results, and also the authors did not claim all the obtained results. 

Author Response

Response to Reviewer 2

Comments and Suggestions for Authors

Q: The research topic is relevant and interesting, but some limitations were found that could be addressed. I would also suggest that the manuscript should meet a professional English editing company.  

A: Thank you very much for reviewer’s comments on our manuscript. Our manuscript has undergone English language editing by MDPI. We hope our manuscript will be devoid of grammatical mistakes and clearly describe our research.

Q: Abstract:

The research theme is not clear in the abstract; What are the market value and opportunities for velvet antlers and why do the authors design this study? Please highlight the clear and concise theme in the abstract.

A: The abstract has been revised based on the reviewers' comments.

Q: Line; 19;  ( In an effort), modify the term.

A: That sentence has been revised (Line 24).

Q: Line; 24; (which were associated to the growth and development of velvet antlers); what is the difference between growth and development? Please revise the sentence.

A: That sentence has been revised (Line 29).

Q: Introduction:

The introduction is confusing and lacking in concepts. For example: Line 31-32; It is the only mammalian organ capable of complete regeneration. 

A: That sentence has been revised (Line 39-40).

Q: Line; 38-39; The research theme is not appropriate, revise it.

A: The research theme of this study has been revised (Line 46-48)

Q: Lines; 44 and 49; (in winter; which is) the initials are not correct, grammatically, so revise it.  

A: We think the phrase "In winter" in the manuscript is correct, or we may have misunderstood the reviewer's comments. If there is still a need for revision, please give us detailed comments, thank you! Other sentences have been revised (Line 60-63).

Q: Lines; 58-59; rephrased the sentence and remove the term scholars.

A: That sentence has been revised (Line 74-76)

Q: Line; 65; (in a totally or partially); what does it mean? I would suggest, correcting the sentence and rephrased.

A: For miRNAs, the 2-8 bases at the 5' end are the seed region, which are the most evolutionarily conserved and are reverse complementary to the 3'UTR of the target genes. If all 7 bases in the seed region are complementary to the mRNA 3'UTR, it is called complete/total complementary. If only 6 or fewer bases of the seed region are complementary to the target site of the mRNA 3'UTR, it is called partially or almost fully complementary. That sentence has been revised (Line 85-86)

Q: Material and methods:

2.1. Sample Collection

I did not see any ethical statement here. How do they care for the experimental animals, were any necessary measures taken to keep the experimental animals as pain-free as possible? What was the most effective and reliable anesthesia for the removal of antlers and how do administer the anesthesia?

A: In this study, all experimental protocols were approved by the Institutional Animal Care and Use Committee of Qinghai University (Xining, China), and all methods were carried out to approved guidelines and regulations (Code: SL-2022024). “Ethics approval” was added after the “Conclusion” section. Antlers were collected after anaesthetising deer with special Mian Naining (anaesthetic, No.9812, People's Liberation Army (PLA) Military Supplies University Research Institute, China), and hemostasis measures were taken immediately after velvet cutting. After recovering from anaesthesia, deer were returned to the herd. The anesthesia and surgical procedures were conducted by a professional veterinarian.

Q: Line; 98; how do the contaminants are removed from the extracted RNS?

A: There would be residual DNA in the process of RNA extraction. In this study, the extracted RNAs were treated with DNase I to degrade the DNAs and improve the purity of the sample RNAs.

Q: Lines; 118-119; The sentence should be excluded from the method section.

A: That sentence has been removed from the “Materials and Methods” section (Line 138-139).

Q: Lines; 122-123; (for sequences that could not be aligned to the miR-123 Base database but could be mapped to the red deer genome); what does it mean?

A: Novel miRNAs are miRNAs that are not yet stored in the miRBase database. Novel miRNAs are identified using miRDeep2 software based on the distribution information of reads on the pre-miRNA sequences. One of the characteristics of miRNAs is that their pre-miRNA sequences have the characteristic hairpin structure. The pre-miRNA secondary structure prediction was conducted by RNAfold software, and the sequences used for hairpin structure prediction could firstly be mapped to the red deer reference genome.

Q: Line; 128; (we compared DEMs in a two-way comparison 128 between d30 versus d60, d30 versus d90 and d60 versus d90), Is it a two-way comparison? Please elaborate.

A: In this study, the data from the three sample groups were compared in pairs, and the differentially expressed miRNAs of each comparison group were screened respectively. We have revised that sentence (Line 148-151).

Q: Results:

Figure 3. Venn diagram of differentially expressed miRNAs. Please explain the legends comprehensively.

I did not see the Hierarchical diagram of differentially expressed miRNAs.  

A: The legends for each figure have all been supplemented. The Hierarchical diagram of differentially expressed miRNAs was not shown in this study.

Q: Figure 6: how can we compare the validation results to the sequencing results, because the manuscript does not have the graph for Seq data e.g., (seq data vs RTqPCR)…

A: The seven differentially expressed miRNAs used for qPCR detection are among the differentially expressed miRNAs identified by RNA-seq and displayed in Table S1. The qPCR data are compared with RNA-seq data without showing the RNA-seq data of the 7 selected differentially expressed miRNAs again.

Q: Discussion:

The discussion is too short and repetition of information found in the introduction and results, and also the authors did not claim all the obtained results. 

A: We are sorry that we did not discuss our results completely. We have supplemented and revised the “Discussion” section, hoping to make it more complete (Line 391-418).

Thank you very much for the reviewers' comments on our manuscript! We have revised the manuscript and hope to make it more rigorous and scientific.

Round 2

Reviewer 2 Report

Please make the figure 3 as penal figure and label it properly e.g., Figure 3 A  and Figure 3 B. And Also treat the Figure 6 similarly.  

Author Response

Response to Reviewer 2 (Round 2)

Comments and Suggestions for Authors

Q: Please make the figure 3 as penal figure and label it properly e.g., Figure 3 A and Figure 3 B. And Also treat the Figure 6 similarly.

A: Many thanks to the reviewer for reviewing our manuscript for the second round. We have revised Figures 3 and 6 to penal figures based on the reviewers' comments and inserted them into the main text of our manuscript.
